

# The sensitivity and specificity of methylene blue spray to identify the parathyroid gland during thyroidectomy

Patorn Piromchai[1], Thipphailin Juengtrakool[1], Supawan Laohasiriwong[1], Pornthep Kasemsiri[1] and Piti Ungarereevittaya[2]

[1] Department of Otorhinolaryngology, Faculty of Medicine, Khon Kaen University, Khon Kaen, Thailand
[2] Department of Pathology, Faculty of Medicine, Khon Kaen University, Khon Kaen, Thailand

## ABSTRACT

**Background:** Hypocalcemia is a common complication of thyroidectomy resulting from an injury to the parathyroid gland. Methylene blue, which is a medication and dye that has been used for more than a century, is safe and readily available. The previous study has found that methylene blue spray on the surgical field is absorbed by the parathyroid gland faster than in the perithyroidal area. This study was aimed to evaluate the diagnostic value of methylene blue spray to identify the parathyroid gland during thyroid lobectomy.

**Methods:** Patients who underwent thyroid lobectomy were recruited. After the recurrent laryngeal nerve was identified, methylene blue was sprayed onto the thyroid bed. After 5 min, the thyroid bed was inspected for areas in which the blue color had been rapidly absorbed. Biopsies were conducted for histopathology at both the stained area and the area in which the color had faded. The sensitivity, specificity, positive predictive value (PPV), and negative predictive value (NPV) were calculated.

**Results:** A total of 47 patients participated in this study. The sensitivity of methylene blue spray to identify the parathyroid gland during thyroid lobectomy was 92.31% (95% CI [63.97–99.81]) and specificity was 56.79% (95% CI [45.31–67.76]). The PPV was 25.53% (95% CI [20.34–31.53]) and NPV was 97.87% (95% CI [87.39–99.67]). There were no patients with post-operative hypocalcemia, allergic reactions to the methylene blue, or methylene blue toxicity.

**Conclusion:** The methylene blue spray could serve as a screening tool for identification of the parathyroid gland.

# INTRODUCTION

Post-operative hypocalcemia is the most frequently encountered complication after thyroidectomy. The incidence of transient hypocalcemia has been shown to be 7–36%, and that of permanent hypocalcemia to be 15% post thyroidectomy (*Pattou et al., 1998*).

Corresponding author
Patorn Piromchai, patorn@kku.ac.th

Hypocalcemia after thyroidectomy is caused by direct injury to the parathyroid gland or injury to its blood supplies (*Reeve & Thompson, 2000*). Post-operative hypocalcemia affects the physical and psychological health of the patient and lengthens hospital stay. Hypocalcemia can be present with peri-oral numbness, numbness of the fingertips and positive of Chvostek's sign. There may be muscle spasms, cramping, seizures, or cardiac arrhythmia in severe cases. Treatment of hypocalcemia consists of giving patients calcium supplements and vitamin D (*Falk, Birken & Baran, 1988*).

Current guidelines for prevention of parathyroid thyroid injury during thyroid surgery is by anatomically locating the parathyroid gland. However, this gland is difficult to separate from the surrounding fat and lymph nodes, which is a reason that post-operative hypocalcemia still occurs.

Other methods to identify the parathyroid gland during thyroidectomy include a partial biopsy of the gland for pathological examination (*Anton & Wheeler, 2005*), intravenous methylene blue injection (*Bewick & Pfleiderer, 2014*), computerized tomography during thyroid surgery (*Sommerey et al., 2015*), parathyroid specific luminescence (*Suzuki, Numata & Shibuya, 2011*), fine needle aspiration for an analysis of parathyroid hormone levels (*Huang et al., 2013*).

Methylene blue, which is a medication and dye that has been used more than a century, is safe and readily available. A previous case series found that methylene blue spray on the surgical field was absorbed by the parathyroid gland faster than in the surrounding perithyroidal area (*Sari et al., 2012*). This study was aimed at evaluating the diagnostic value of methylene blue spray in identifying the parathyroid gland during thyroidectomy.

# MATERIALS AND METHODS

## Study design and setting

This prospective diagnostic study was conducted from July 2016 to April 2018 at the Khon Kaen University, Faculty of Medicine, Department of Otorhinolaryngology, Thailand.

The inclusion criteria were patients aged 18 years or older and underwent thyroid lobectomy. Exclusion criteria were previous thyroid surgery, methylene blue allergies, renal diseases, G6PD deficiency, pregnancy, and having received monoamine oxidase inhibitors such as tranylcypromine, isocarboxazid, phenelzine, furazolidone, isoniazid, procarbazine, or linezolid.

Recurrent laryngeal nerve function, thyroid hormone levels, and calcium levels were pre- and post-operatively evaluated. All participants underwent open thyroidectomy under general anesthesia.

During thyroidectomy, the thyroid gland was pulled back to identify the recurrent laryngeal nerve and upper and lower parathyroid glands. One ml (10 mg) of 1% methylene blue solution was sprayed over the thyroid bed and perithyroidal tissue (Fig. 1). A 5 min after spraying, parathyroid glands had absorbed the blue stain and regained their original yellow color. Other tissues were still stained blue (Figs. 1–3).

The biopsies were conducted at (1) the area that was suspected to be the parathyroid gland based on surgical anatomy, which had regained its yellow color and (2) the

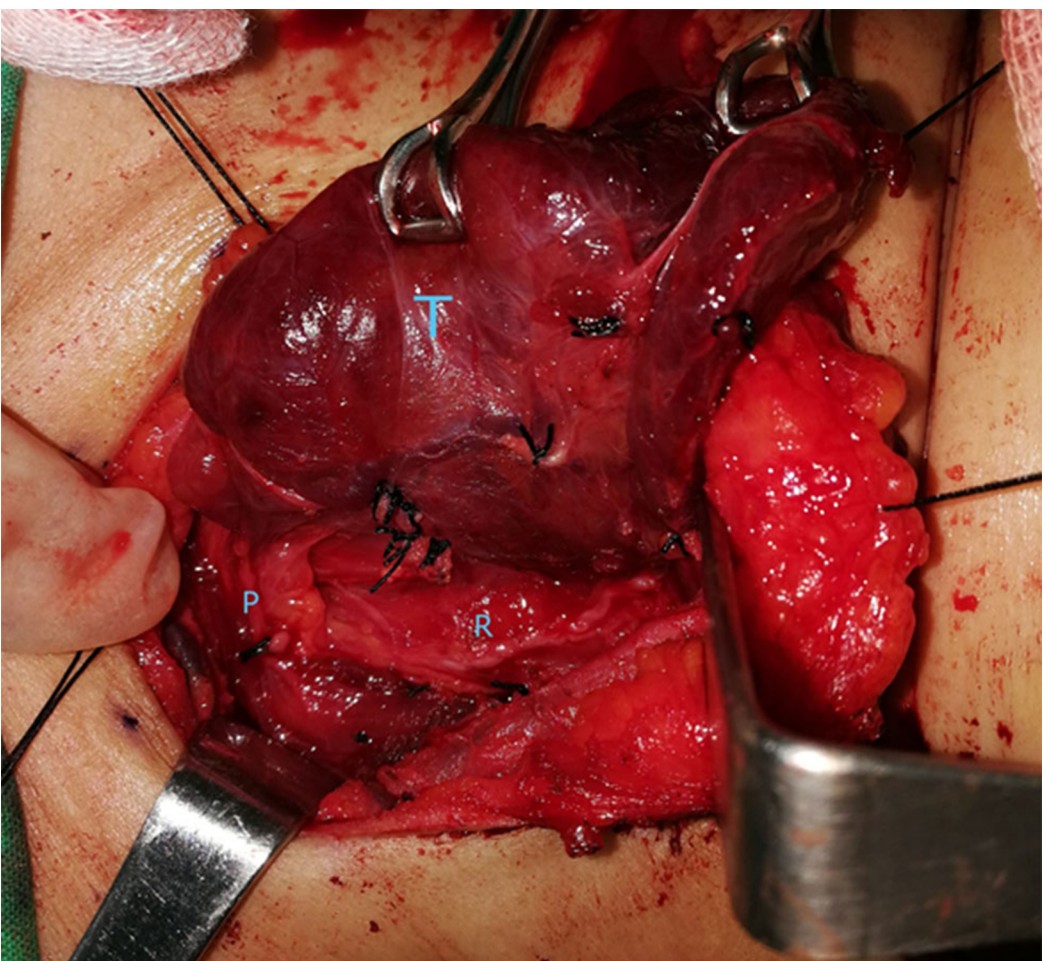

**Figure 1 Before spraying of methylene blue.** T, thyroid gland; R, recurrent laryngeal nerve; P, parathyroid gland. Photo credit: Patorn Piromchai, MD, MSc, PhD, FRCOT, FICS.

blue-stained area that was not suspected to be the parathyroid gland. An incisional biopsy of two mm diameter for each specimen was performed carefully to prevent an injury to a recurrent laryngeal nerve or thyroid gland by consultant otolaryngologists or under their supervision.

The two specimens from each patient were submitted for histopathology. To blind the pathologist, a computer-generated randomization list was used. The specimens were randomly labeled as either specimen "A" or "B."

The patients received standard post-operative care. The post-operative complications and duration of hospital stay were recorded.

## Statistical analysis

The sample size was calculated using an estimated sensitivity of 70% with a range of sensitivity from 60% to 80%. With a significance level of 0.5 and power of 90%, the total number of specimens required were determined to be 81.

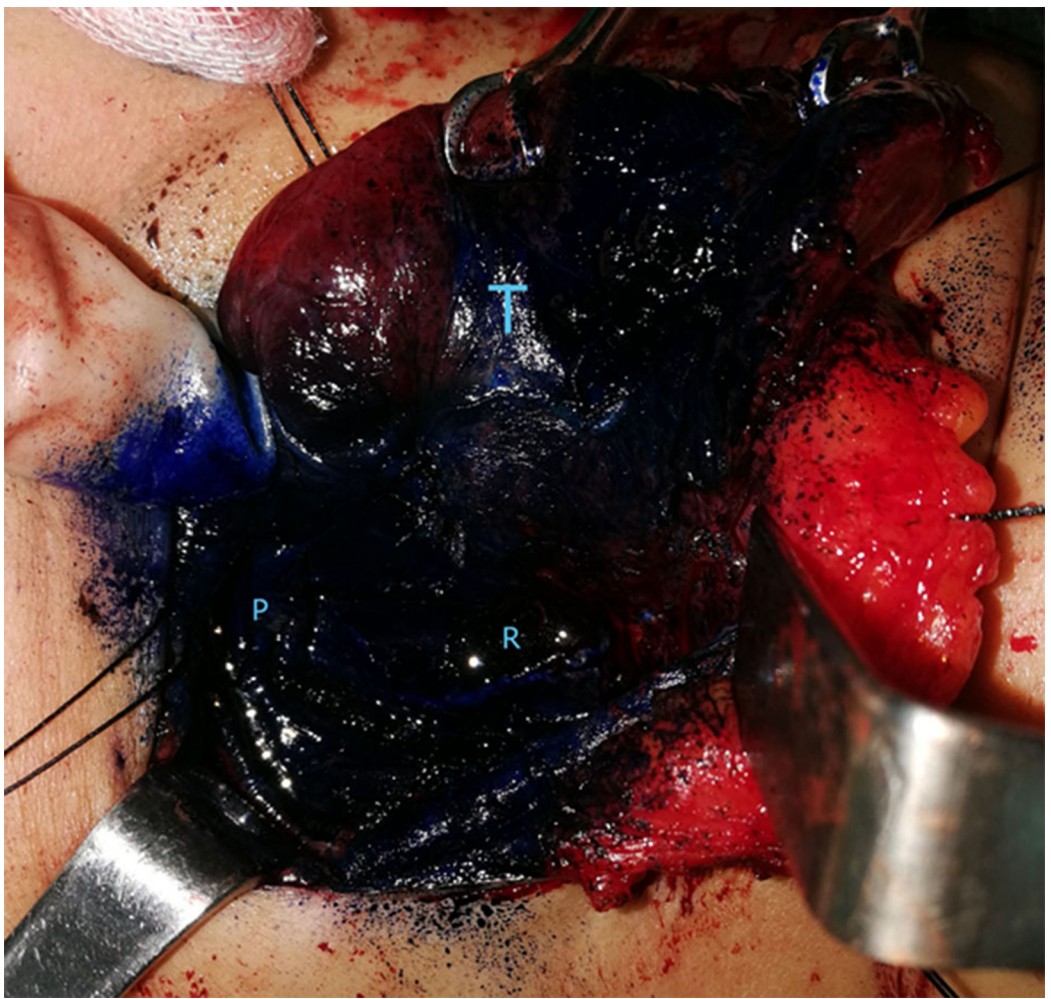

**Figure 2** **Immediately after spraying of methylene blue.** T, thyroid gland; R, recurrent laryngeal nerve; P, parathyroid gland. Photo credit: Patorn Piromchai, MD, MSc, PhD, FRCOT, FICS.

The categorical variables were presented in the form of frequencies and percentages. The associations among categorical variables were assessed using a chi-square test. The continuous variables were presented in the form of means. The sensitivity, specificity, positive predictive value (PPV), and negative predictive value (NPV) were calculated with a 95% confident interval.

## Ethical considerations

This study was approved by the Khon Kaen University Ethics Committee in Human Research (HE591169) and registered in the Thai Clinical Trials Registry (TCTR20160726003). Written informed consent to participate in this study was provided by all patients enrolled.

## RESULTS

A total of 47 patients participated in this study, six of whom were male (13.33%) and 41 of whom were female (86.67%). The average age of the patients was 50.30 years old

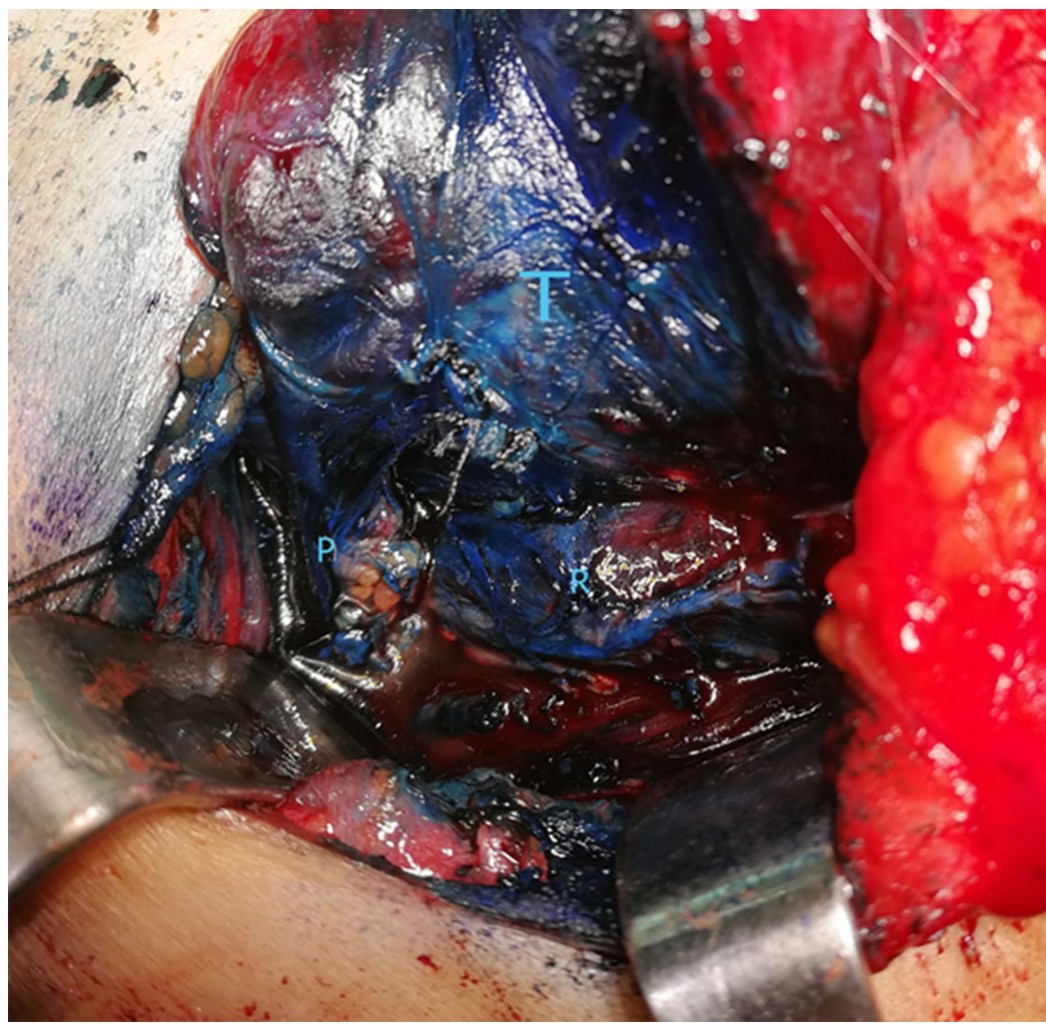

**Figure 3 A 5 min after spraying of methylene blue.** T, thyroid gland; R, recurrent laryngeal nerve; P, parathyroid gland. Photo credit: Patorn Piromchai, MD, MSc, PhD, FRCOT, FICS.

(ranging from 22 to 81 years old). Pre-operative diagnoses were as follows: unilateral nontoxic single thyroid nodule ($n = 43$), thyroid toxic adenoma ($n = 1$), Grave's disease with cold nodule ($n = 3$). The cytology from fine needle aspiration were non-diagnosis ($n = 19$), benign follicular nodule ($n = 17$), follicular cell neoplasm ($n = 3$), and atypia of undetermined significance ($n = 8$). The size of thyroid nodule according to physical examination was between 1 and 10 cm (average $3.83 \pm 1.77$ cm) (Table 1).

A total of 12 of the 47 specimens that were positive for rapid wash-out were confirmed to be parathyroid tissue by histopathology. A total of 34 were fibrofatty tissue, and one was a lymph node. Conversely, only one of the 47 negative specimens were found to be parathyroid tissue, while the remaining 46 were found to be fibrofatty tissue (Table 2).

Sensitivity of methylene blue spray to identify the parathyroid gland during thyroidectomy was 92.31% (95% CI [63.97–99.81]) and specificity was 56.79%

**Table 1 Patient characteristics.**

| Characteristics | | N |
|---|---|---|
| Age (min–max) | | 50 (22–81) |
| Male | | 6 (12.77%) |
| Female | | 41 (87.23%) |
| Pre-operative diagnosis | Nontoxic single thyroid nodule | 43 (91.15%) |
| | Grave's disease with cold nodule | 3 (6.38%) |
| | Thyroid toxic adenoma | 1 (2.13%) |
| Pre-operative cytology | Non-diagnostic | 19 (39.60%) |
| | Benign follicular nodule | 17 (35.40%) |
| | Follicular cell neoplasm | 3 (6.30%) |
| | Atypia of undetermined significant | 8 (16.70%) |
| Size of thyroid nodule (cm) | | 3.83 ± 1.77 |

**Table 2 Diagnostic value of methylene blue spray.**

| 5 min after methylene blue spray | Histopathology | |
|---|---|---|
| | Positive for parathyroid gland ($n = 13$) | Negative for parathyroid gland ($n = 81$) |
| Rapid wash-out ($n = 47$) | 12 | 35 |
| Stained blue ($n = 47$) | 1 | 46 |

(95% CI [45.31–67.76]). The PPV was 25.53% (95% CI [20.34–31.53]) and NPV was 97.87% (95% CI [87.39–99.67]).

Histopathology results of the thyroid nodules were as follows: follicular adenoma ($n = 38$), papillary thyroid carcinoma ($n = 3$), Hurthle cell adenoma ($n = 4$), degenerative cyst ($n = 1$), and chronic thyroiditis ($n = 1$). Patients were discharged after an average of 3.38 days (3–5 days) with no major complications (no signs or symptoms of post-operative hypocalcemia, no post-operative vocal cord immobility, and no post-operative hematoma). None of the patients had allergic reactions to the methylene blue or methylene blue toxicity.

## DISCUSSION

There have been various methods proposed to identify the parathyroid gland in order to decrease the incidence of post-operative hypocalcemia. We summarized the advantages and disadvantages of each method in Table 3.

Methylene blue is certified by the US Food and Drug Administration. It can be used in a variety of medical treatments, including hereditary methemoglobinemia, acute acquired methemoglobinemia, prevention of urinary tract infection in the elderly, and localization of nerves and endocrine tissues. There are few side effects associated with methylene blue. However, toxicity can occur if more than five mg/kg is used. Symptoms of toxicity include dizziness, nausea, vomiting, headache, abdominal pain, and confusion (*Schirmer et al., 2011*; *Oz et al., 2011*; *Gillman, 2006*).

Table 3 Methods to identify the parathyroid gland.

| Methods | Advantages | Disadvantages |
|---|---|---|
| 1. Frozen section (*Anton & Wheeler, 2005*) | – Close to gold standard. <br> – Accuracy more than 99%. | – Used only in parathyroid diseases. |
| 2. Surgical anatomy of the superior and inferior thyroid arteries (*Kapre, 2009*) | – Low cost. | – Post-operative hypoparathyroidism still occurs in around 10% of cases. |
| 3. Intraoperative optical coherence tomography imaging (*Sommerey et al., 2015*) | | – An accurate differentiation between parathyroid tissue and lymph nodes was not possible. <br> – High cost. |
| 4. Intravenous methylene blue injection (*Bewick & Pfleiderer, 2014*) | – Low cost. | – 78.6% of cases stained positively with methylene blue. <br> – 5.8% of patients suffer from a systemic complication of methylene blue injection. |
| 5. Intraoperative photodynamic detection of normal parathyroid glands using 5-ALA (*Suzuki, Numata & Shibuya, 2011*) | – 100% specificity. | – High cost, long test duration and not available in all hospitals. <br> – Side effects: four patients had nausea and two had to vomit. |
| 6. FNA with measurement of parathyroid hormone levels in thyroidectomy (*Huang et al., 2013*) | – 97.8% sensitivity and 100% specificity. | – High cost and not available in all hospitals. |
| 7. Intraoperative methylene blue spray | – 92.31% sensitivity. <br> – Safe, rapid, easy, and low cost. | |

*Sari et al. (2012)* studied 56 patients who underwent thyroidectomy in which methylene blue spray was used to locate the parathyroid gland. They found that the parathyroid gland was able to absorb the blue staining and regain its original yellow color in 3 min, while other tissues took longer. They found that the thyroid took 15 min and fat, tendon, and muscle took more than 25 min. They proposed the theory that parathyroid glands can absorb methylene blue faster than other tissues because they have a dense lymphovascular pattern.

In this study, we further evaluated the diagnostic value of methylene blue spray. The sensitivity of spraying methylene blue was 92.31% (95% CI [63.97–99.81]) and specificity was 56.79% (95% CI [45.31–67.76]). The PPV was 25.53% (95% CI [20.34–31.53]) and NPV was 97.87% (95% CI [87.39–99.67]). Our result found that methylene blue spray at thyroid bed to identify the parathyroid gland was highly sensitive and suitable to use as a screening tool to avoid injury to the parathyroid gland.

To achieve this high sensitivity, the surgeon needs to identify the area suspected to be parathyroid gland first and confirms by methylene blue spray. The solely use of methylene blue spray without surgical skills and knowledge was highly not recommended.

In all of our cases, we can observe the difference of blue shading between the suspected area and another area. However, in an unexpected situation, the surgeon may feel uncertain of the shading. In this case, we recommended using an additional tool such as fine-needle aspiration (FNA) with measurement of parathyroid hormone levels to identify parathyroid gland.

To our knowledge, this is the first diagnostic study to evaluate the value of methylene blue spray to identify parathyroid gland. The results of this study can be applied in order to help to identify the parathyroid gland during thyroidectomy and may decrease post-operative hypocalcemia.

## CONCLUSIONS

The methylene blue spray could serve as a screening tool for identification of the parathyroid gland.

## ACKNOWLEDGEMENTS

The authors would like to thank the staff and nurses at Srinagarind Hospital for their excellent care of the patients.

### Funding

The authors received internal funding from the Faculty of Medicine, Khon Kaen University (Grant Number IN59342). The funders had no role in study design, data collection and analysis, decision to publish, or preparation of the manuscript.

### Grant Disclosure

The following grant information was disclosed by the authors:
Faculty of Medicine, Khon Kaen University: IN59342.

### Competing Interests

The authors declare that they have no competing interests.

### Author Contributions

- Patorn Piromchai conceived and designed the experiments, performed the experiments, analyzed the data, contributed reagents/materials/analysis tools, prepared figures and/or tables, authored or reviewed drafts of the paper, approved the final draft.

- Thipphailin Juengtrakool conceived and designed the experiments, performed the experiments, analyzed the data, contributed reagents/materials/analysis tools, prepared figures and/or tables, authored or reviewed drafts of the paper, approved the final draft.
- Supawan Laohasiriwong conceived and designed the experiments, performed the experiments, approved the final draft.
- Pornthep Kasemsiri performed the experiments, approved the final draft.
- Piti Ungarereevittaya performed the experiments, approved the final draft.

## Human Ethics

The following information was supplied relating to ethical approvals (i.e., approving body and any reference numbers):

The Khon Kaen University granted Ethical approval to carry out the study within its facilities (Ethical Application Ref: HE591169).

## Data Availability

The raw data is available in the Supplemental File.

## Supplemental Information

Supplemental information for this article can be found online at http://dx.doi.org/10.7717/peerj.6376#supplemental-information.

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
