# Peer review of "The sensitivity and specificity of methylene blue spray to identify the parathyroid gland during thyroidectomy"

_PeerJ, doi:10.7717/peerj.6376_

## Round 0.1 · original submission · Minor Revisions

Dear authors,

Your paper has been reviewed by two experts in the topic and they have indicated some changes which you must address in a revised version of the text.

With respect and kind regards,
Dr Palazón-Bru (academic editor for PeerJ)

·

Basic reporting

This is a well-written study with clear professional English. It is well-structured with a reasonable rationale. Literature has been reviewed. Figures have high quality.

Experimental design

Research methodology is adequate with sufficient detail. Research question is interesting. Knowledge gap has been identified.

Validity of the findings

Data is valid. Clinical benefit is meaningful and has been stated.

Additional comments

The authors sprayed methylene blue onto the thyroid bed of 47 patients receiving thyroid lobectomy. The area with faded color was suspected to be parathyroid glands. Histopathology was a reference. Sensitivity, specificity, PPV and NPV were analyzed. I reckon the findings from this study are clinical meaningful and very useful in practice.
1. Methylene blue was sprayed after the recurrent laryngeal nerve and upper and lower parathyroid glands were identified and the thyroid gland was pulled back (Line 78-80). While I believe the high sensitivity of methylene blue will be very useful, this study has shown us that methylene blue may not be necessary if surgeons have high surgical skill and good knowledge on surgical anatomy. Please discuss.
2. The biopsies were conducted at the area that was suspected to be the parathyroid gland based on surgical anatomy, which had regained its yellow color (Line 83-84). Please report if there was any case and any area which was suspected by surgical anatomy but having blue-stained color. I assume that there were none. Please confirm.

Reviewer 2 ·

Basic reporting

It was study that was clearly written and could be easily understood. There was a clear rational. The objective of the study was clear. The study report was easily to understand and there were also high-quality figure.

Experimental design

Research methodology could prove the research question of the study very well. It was a study with low risk bias with good quality.

Validity of the findings

Data were valid and could really be used in the clinical practice.

Additional comments

This is a prospective diagnostic study which the authors sprayed methylene blue onto the thyroid bed of 47 patients receiving thyroid lobectomy. After five minutes, Biopsies were conducted for histopathology at both the stained area and the area in which the color had faded. The area with faded color was suspected to be parathyroid glands. Histopathology was a reference. The sensitivity, specificity, positive predictive value (PPV), and negative predictive value (NPV) were calculated.
1. If after spraying methylene blue, the difference of the area that can quickly absorb the color and other area could not be observed. Where to make the biopsy or to find parathyroid according to surgical anatomy knowledge. Please discuss.
2. Using of methylene blue spray is a test with high level of sensitivity but the specificity is not very good so it is suitable to be a screening tool. Therefore, other tests with good specificity should be used along. The conclusion should be toned down.

---

## Round 0.2 · accepted · Accept

Dear authors,

It is a pleasure to accept your paper in its current form in PeerJ.

Congratulations!

Withn respect and kind regards,
Dr Palazón-Bru (academic editor for PeerJ)

# ·

Basic reporting

This is a well-written study with clear professional English. It is well-structured with a reasonable rationale. Literature has been reviewed. Figures have high quality.

Experimental design

Research methodology is adequate with sufficient detail. Research question is interesting. Knowledge gap has been identified.

Validity of the findings

Data is valid. Clinical benefit is meaningful and has been stated.

Additional comments

The authors have made adequate revisions. Thank you.

Reviewer 2 ·

Basic reporting

Thank you for your revisions and thoughtful work. I have no further comments.

Experimental design

Thank you for your revisions and thoughtful work. I have no further comments.

Validity of the findings

Thank you for your revisions and thoughtful work. I have no further comments.

Additional comments

Thank you for your revisions and thoughtful work. I have no further comments.